# Does hand proximity enhance letter identification?

**Giordana Grossi[1], Annie J. Olmstead[2]\*, Danielle Lukaszewski[1]**

**1** Department of Psychology, State University of New York at New Paltz, New Paltz, New York, United States of America, **2** Department of Communication Sciences and Disorders, The Pennsylvania State University, University Park, Pennsylvania, United States of America

\* ajo150@psu.edu

## Abstract

Adam et al. (2012) found that letters were identified more accurately when presented near, compared to away from, the hands. Participants performed the task in two conditions: with their hands held stationary and with their hands moving towards and away from the target letters. The near-hands effect included the contribution of both static and dynamic trials. Further studies showed that accuracy in letter discrimination was higher when hands were away from a target (a far-hands effect) and moving toward it, suggesting an interaction between hand position and movement direction. The present study aimed to test whether hand proximity affects letter identification when the hands are stationary, as it remains unclear if this effect can be reliably observed. Participants viewed strings of three consonants, briefly presented and masked, and had to verbally report their identity. Stimuli were presented under two different hand conditions: proximal and distal. The predicted effects of letter position and stimulus duration were all statistically significant and robust; however, we did not observe a hand proximity effect.

**Data Availability Statement:** All relevant data are within the manuscript and its Supporting Information files.

**Funding:** The author(s) received no specific funding for this work.

## Introduction

Hand proximity (or near-hands) effects have been observed in a variety of experimental paradigms [for reviews, see 1–3]. In the typical set up for these studies, participants perform a task in which the stimuli are presented near (proximal condition) or away (distal condition) from the hands, with the order of these two conditions counterbalanced across participants. Depending on the task, hand proximity has been found to enhance or impoverish the processing of visual stimuli, either in terms of reaction time (RT) or accuracy.

The interpretation of these effects has varied. Initially, these effects were explained in terms of increased attentional resources for items near the hands; the postulated mechanism was the activation of bimodal neurons in parietal cortex when visual stimuli are presented near the hands, potentially relevant for action [e.g., 4, 5]. However, decreased performance near the hands was reported in some studies [e.g., 6, 7]. Subsequent hypotheses were proposed to explain results in specific contexts; hypotheses ranged from enhanced cognitive control near the hands [8] to differential effects depending on whether the task engaged the magnocellular

**Competing interests:** The authors have declared that no competing interests exist.

or parvocellular pathways [2, 3, 9] or involved a specific grasping position [10]. This complexity has invited some authors to rethink the way these effects are investigated. For example, Gozli and Deng [11] argued that near-hands effects cannot be explained by a single mechanism and that hand proximity cannot be conceptualized as an "isolable factor," one that would affect visual perception in the same way regardless of the nature of the task (see their article for an in-depth analysis). According to Gozli and Deng [11, p12], while a change of strategy is needed to study hand proximity effects and their theoretical contributions, their study could be of "practical relevance" if the focus of investigation involves "intrinsically important" activities.

Studies on hand proximity are indeed sometimes framed in terms of the implications of near-hands effects in realistic situations. One of these activities is reading. For example, in light of research suggesting enhanced spatial processing near the hands, Davoli and colleagues [7] investigated whether hand proximity also affected semantic processing during reading. They adopted two tasks: in the first, participants were asked to decide whether or not English sentences were semantically appropriate and therefore acceptable; in the second, participants identified the color of font with congruent (e.g., RED in red) and incongruent (e.g., RED in green) words (the Stroop task). The authors found that hand proximity modulated the acceptability of the sentences and the Stroop effect, a modulation that the authors interpreted as an impoverishment of semantic processing near the hands. At the end of their paper, the authors discuss the potential tradeoff that readers might want to consider when reading, as hand proximity enhances spatial processing but also negatively impacts semantic processing.

However, in four different experiments, Grossi, Olmstead, and Stoudt [12] failed to observe hand proximity effects on semantic processing. The authors employed a variety of tasks (sentence decision task, semantic categorization, picture naming) and observed classic behavioral effects of semantic processing, but not near-hands effects. In addition, Grossi and colleagues remarked that the tradeoff discussed by Davoli et al. [7] was only assumed to exist, not demonstrated. Therefore, the suggestion that hand proximity affects reading was premature.

Another study has investigated near-hands effects on processes important to reading, more specifically, letter identification. Adam and colleagues [13] aimed to establish whether evidence for enhanced visual analysis near the hands could be found by using accuracy, rather than the more typical RT as a dependent variable. Their reasoning was that effects on RT might reflect the role of response-related variables and noise due to the compatibility between stimulus and response in lateralized trials, rather than the evaluation of the stimulus itself [13]. In their first experiment, participants viewed unpronounceable trigrams presented in the center of a computer screen with variable durations (27, 40, 53, 66, and 80 ms) and masked. Participants were asked to report the identity of the letters in the order they wanted. They performed this task in two conditions (see their Fig 1, p. 1535): in the static one (typical in hand proximity experiments), participants kept their hand on keypads, positioned under the monitor, at three different distances from the central stimuli (0 cm, 17.5 cm, or 35 cm, or near, intermediate, and far distance, respectively); in the dynamic condition, they moved the keypads with their hands, mirror-like, towards or away from the central letters by following a continuous and self-paced motion. In the latter condition, trigrams were presented between 3 and 4 seconds from each other; the position of the hands was determined offline and measured in a way that allowed for having an equal number of trials in the three hand positions (0–9 cm, 9–26 cm, 29–44 cm). Besides the expected significant effects of stimulus duration, letter position, and their interaction, accuracy was higher for stimuli presented near the hands than for stimuli presented in the intermediate and far positions. The main effect of hand position did not interact with hand condition (static vs. dynamic); therefore, both static and dynamic trials contributed to it. These findings were replicated in a second, and more challenging,

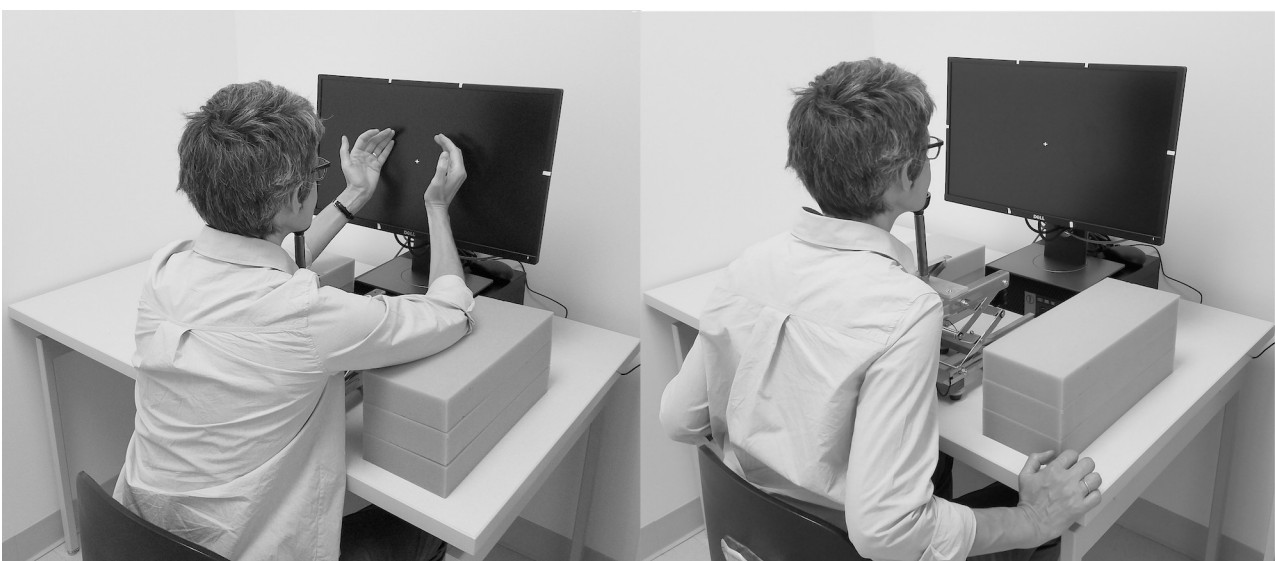

**Fig 1. Experimental set-up.** The setup used for the proximal (left) and distal (right) position used in the experiment.

experiment in which participants were presented with a display of six letters, half written in white and half in red; their task was to report the three letters written in white. Adam and colleagues interpreted these findings as reflecting "a facilitative effect on perceptual encoding processes, which fits with the suggestion that the enhancement of vision for near-hand objects reflects a mechanism that facilitates the detailed evaluation of objects for potential interaction (Abrams et al., 2008)." (p. 1537). The fact that the effect was found with the hands moving was interpreted within the framework of a parietal bimodal neuron systems aimed to organize action within the peripersonal space [13].

Adam and colleagues' findings might be important in terms of practical relevance, as books and other reading materials are held by stationary hands. However, because the hand proximity effect in their study included both static and dynamic trials, it remains unclear if the effect can be found when the hands do not move. Two further studies were conducted to clarify the contribution of hand movement on perceptual processing. Festman, Adam, Pratt, and Fischer [14] asked participants to discriminate between two rotated letters while moving their right hand back and forth under a computer monitor (hence, the hand was not visible, as in Adam et al. [13]). Letters were briefly presented and masked on the right or left of a central fixation point while participants moved their hand and passed three critical positions (left, center, right; see their Fig 1, p. 646). After a movement completion (e.g., right to left), the two stimuli were presented near the fixation point; participants selected which letter had just been presented with a mouse click. The authors were thus able to test the effect of hand position (near, intermediate, or far) on letter discrimination and whether the direction of the hand movement (toward or away from the target letter) had an impact. The authors found that direction of the hand movement interacted with hand proximity (which was not significant as a main effect); more specifically, when the hand moved toward the target, letter discrimination was more accurate away from the target (a "far-hands effect"; see their Fig 3C, p. 647); when the hand moved away from the target, letter discrimination was more accurate near compared to far away from the target (this effect was termed "a trend" by the authors, although no follow-up statistical analyses were reported). The authors concluded that the direction of movement biases the allocation of attention; near-hands effect was either absent or overshadowed by the

effect created by the hand movement, which shifted the participants' focus of attention ahead toward an action-relevant location.

In a following study, Festman and colleagues aimed to investigate the relative contribution of near- and far-hands effects [15]. In this study, the participants' left hand was positioned on the left of the screen, so that the letters presented on the left of the fixation point were near the static left hand (see their Fig 1, p. 3). Given the presence of a static left hand, the authors analyzed left and right target trials separately, although the position of the target did not interact with any other factors (p. 3). The interaction between hand proximity and hand movement direction was replicated for targets presented on the right of the fixation but not for those presented on the left, near the static hand. More specifically, participants were more accurate in identifying the right targets when the right hand was far from the target (that is, on the left of the participant's fixation) and moved toward the target compared to when it moved away from it; in contrast, no effects were reported for left targets (for reference, see their Fig 3C and 3D, p. 4). Festman and colleagues concluded that the way spatial attention is deployed is shaped by both static and dynamic aspects of the hands. As the modulation of the far-hands effect was not present for left targets, the authors concluded that the presence of the static left hand erased the far-hand effect for targets presented on the left (for a discussion and similar interpretation of these results, see [16]).

## The current study

The previous studies do not offer a clear conclusion on whether letter identification is affected by hand proximity when the hands are stationary. In Adam et al. [13], letter processing was enhanced when letters were presented near the hands, but the main effect included the contribution of both static and dynamic trials. Using a discrimination task, Festman and colleagues were unable to replicate this effect and instead observed a far-hands effect when the hands were moving towards the targets [14, 15]. In all three studies, the assessment of hand proximity effect was thus contaminated by the presence of the hands moving, statistically or procedurally (stimuli were presented while the hands moved). Festman and colleagues [15] invoked the presence of near-hands effects to explain the discrepancy of results between left and right trials (no near-hands effect found for letters presented on the left of fixation, near a static left hand, purportedly due to the erasure of effect for left targets by the static left hand). This interpretation should be considered with caution for two reasons: first, no higher accuracy for left targets was observed when the right hand was near the left hand; second, the authors analyzed left and right targets separately, although the position of the target did not the interact with the hand proximity and hand movement. The lack of a three-way interaction therefore complicates the interpretation of the results and the conclusion that different mechanisms were at play for left and right targets.

For these reasons, it remains unclear whether hand proximity enhances letter identification and therefore the early stages of reading. Indeed, some authors have questioned the existence of the effect in Adam and colleagues' 2012 study (e.g., [2]). Thus, it is necessary to establish whether the effect can be observed in conditions in which movement of the hands is not a feature of the task. In the current study, letters were presented to participants such that their hands were visible and positioned near and away from the stimulus; in addition, the hands in the proximal position surrounded the targets, slightly facing each other as might be typical in an everyday reading context.

Based on Adam et al. [13], we made the following predictions:

- Main effect of Stimulus Duration: Accuracy would be higher for longer, compared to shorter, presentation times.

- Main effect of Letter Position: Accuracy would be higher for letters in the leftmost position compared to the other two positions.

- Letter position x Stimulus Duration interaction: The advantage of the leftmost letter would decrease with increasing presentation time.

- If hand proximity improves letter identification, accuracy would be higher in the proximal compared to the distal position (main effect of Hand Proximity).

## Materials and methods

### Participants

A power analysis conducted with G*Power 3.1.9.4 and based on the effect reported by Adam and colleagues ($\eta^2 = 0.22$ in Experiment 1) showed that a sample size of 43 participants was required to obtain 80% power with a 0.5 uncertainty adjustment. Forty-three students from SUNY New Paltz volunteered for this study through the Psychology Department subject pool. Participants (31 self-identified as female, 11 as male, 1 as other) had an average age of 20.8 years (range = 18–30 years) and normal or corrected-to-normal vision. Based on self-report, they were native English speakers and did not have a history of neurological disorders, learning disabilities, or dyslexia. Participants received subject pool credits for their participation. The study was approved by the Institutional Review Board at SUNY New Paltz. Informed consent was obtained by all participants in written form.

### Apparatus

Participants faced a 21.5-in. flatscreen monitor at a distance of 40 cm, which was maintained using a chinrest. In the proximal condition, participants rested their arms on two sets of stackable foam pillows while their hands surrounded the stimuli, 7.6 cm on each side; their palms slightly faced the stimuli, so not to block the room light (Fig 1). In the distal position, they kept their hands on the table, aligned with their body, 35 cm on each side. The experiment was run with the software E-Prime.

**Stimuli.** Stimuli were comprised of three different uppercase letters, presented in white Courier New font (size 20) against a black background. Following Adam and colleagues, letters were selected from all consonants with the exception of "Y" and "M", for a total of 19 different letters. Two lists, each including 152 stimuli (38 triplets X 4 presentation times) included different sets of stimuli. In each list, each letter appeared eight times in the first, second, and third position. Each list was presented in a different hand position for each participant for a total of 304 trials (152 stimuli x 2 hand positions). The individual letters measured approximately 6 × 5 mm and were separated by 6–7 mm. Language experience and neurological history were assessed with a short questionnaire administered before the experimental session.

### Procedure

Participants performed the experiment in a sound-attenuated booth. The sequence of events was the following: a fixation point was presented for 1500 ms and replaced by three letters for either 33, 50, 67, or 83 ms. The letters were then replaced by a masking stimulus (a row of five hash marks) for 200 ms. Participants were asked to name the letters they identified in whatever order they wanted; they were encouraged to guess if necessary. The participants' responses were written on a sheet and then typed into keyboard by the experimenter, who sat at the computer's side and was blind to the identity of the presented stimuli. The typing of the third letter prompted the beginning of the next trial. Block order of presentation (proximal first, distal

first) was counterbalanced across participants (21 participants started in the proximal position, 22 in the distal position). Stimulus list, order of presentation of the stimuli, and the presentation time associated with the stimuli varied randomly across participants. Participants started the experimental session with 12 practice trials, three for each presentation time; none of these triplets were included in two experimental lists. All participants were debriefed and thanked at the end of the experiment.

### Design and analyses

The independent variables manipulated within subjects were the following: Hand Position (distal, proximal), Stimulus Duration (33, 50, 67, 83 ms), and Letter Position (left, middle, right). Accuracy in reporting the identity of letters, regardless of their position, constituted the dependent variable. Accuracies were submitted to a 2 Hand Position X 3 Stimulus Duration X 2 Letter Position repeated measures ANOVA.

### Results

Results are shown in Table 1 and Fig 2 (individual data are available in S1 Dataset). Letter identification accuracy was higher for longer, compared to shorter, presentation times ($F(3,126)$ = 596.11, $p < .0001$, $\eta^2_P$ = .93) and for letters presented in the left position compared to the letters presented in the other two positions ($F(2,84)$ = 131.38, $p < .0001$, $\eta^2_P$ = .76). In addition, the advantage of the left letter decreased with increasing presentation time (Letter Position x Stimulus Duration interaction, $F(6,252)$ = 15.77, $p < .0001$, $\eta^2_P$ = .23). Hand proximity was not significant as a main effect (Proximal, $M$ = 68.76, $SD$ = 22.71; Distal, $M$ = 68.83, $SD$ = 23.63; $F(1,42)$ = 0.006, $p$ = .936) and did not interact with any other factors (all $p$'s>0.60).

In order to evaluate the null effect of hand position, we calculated the Bayes Factor for hand proximity using a Bayesian paired samples t-test analysis on the overall accuracies in the distal and proximal position. This analysis was conducted using SPSS with default priors. The $BF_1$ of

**Table 1. Mean accuracies by condition.**

|  | Stimulus Duration | | | | |
| --- | --- | --- | --- | --- | --- |
| **Proximal** | **33 ms** | **50 ms** | **67 ms** | **83 ms** | **Means** |
| **Left position** | 63.71 (17.32) | 82.93 (10.58) | 89.72 (8.59) | 92.04 (6.88) | 82.10 (15.99) |
| **Middle position** | 34.46 (13.61) | 59.18 (18.59) | 73.68 (16.16) | 80.17 (13.50) | 61.87 (23.44) |
| **Right position** | 39.60 (16.28) | 59.79 (17.87) | 72.09 (16.10) | 77.72 (15.22) | 62.30 (21.89) |
| **Means** | 45.92 (20.25) | 67.30 (19.44) | 78.50 (16.1) | 83.31 (13.81) | |
| **Distal** | **33 ms** | **50 ms** | **67 ms** | **83 ms** | **Means** |
| **Left position** | 65.06 (17.92) | 84.82 (11.71) | 89.60 (9.98) | 92.17 (7.41) | 82.91 (16.26) |
| **Middle position** | 35.07 (15.72) | 58.14 (18.32) | 72.58 (18.70) | 79.07 (16.48) | 61.21 (24.14) |
| **Right position** | 38.19 (13.42) | 60.47 (20.32) | 72.03 (19.40) | 78.83 (15.38) | 62.38 (23.14) |
| **Means** | 46.10 (20.69) | 67.81 (20.91) | 78.07 (18.38) | 83.85 (14.96) | |

Accuracy (in percentages) of correctly identified letters as a function of hand position, stimulus duration, and letter position. Standard deviations are shown in parentheses.

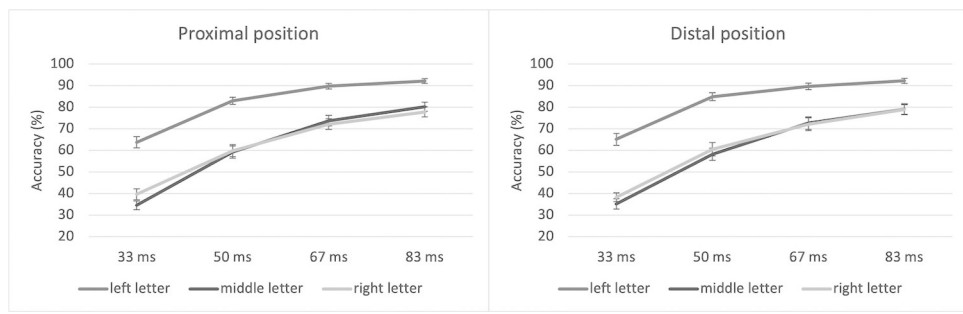

**Fig 2. Accuracy by condition.** Accuracy (in percentages) data in the proximal and distal conditions. Error bars represent 95% confidence intervals.

8.375, corresponding to a $BF_{10} = 0.119$, indicated moderate evidence in favor of the null hypothesis [18].

## Discussion

The current study aimed to investigate whether letter identification is enhanced by hand proximity. Adam and colleagues [13] were the first to suggest that this is the case in an experiment in which participants were tested in two conditions: when the hands were kept statically at three different distances from a central target made of three letters and when the hands moved rhythmically together toward and away from the letters. As both static and dynamic trials contributed to the effect observed by the authors and as following studies reported far-hands (not near-hands) effects tied to the movement of the hands [14, 15], it remained unclear whether hand proximity affects letter identification when the hands are stationary. As mentioned, Festman and colleagues [14] did not report a near-hands effect but suggested that this factor was at play as revealed by interactions with direction of hand movement. However, the absence of follow-up statistical analyses complicates their interpretation. In addition, the differential pattern of effects for left and right target in Festman et al. [15] is difficult to interpret because the target position did not interact with the other factors in the study. Here, we adopted an experimental setup aimed to directly assess near-hands effects on letter identification when hand movement is not part of the task; while we replicated all other effects described by Adam and colleagues (effects of letter position, stimulus duration, and their interaction), we failed to find a near-hands effect. While we cannot establish whether the effects reported by Adams et al. [13] are the result of a Type I error or are dependent on hand movement, our findings do not support the view that letter identification is enhanced near the hands when the hands are stationary.

In their general discussion, Adam and colleagues acknowledge that the effects that they observed were "relatively small in magnitude" and identify some of the factors that might have been responsible for their size. For example, the hands were not visible, facing downward from the stimuli, and positioned underneath the monitor, not at its sides, as in the typical setting for these studies. In our experiment, we aimed to maximize the probability of finding an effect by avoiding these limitations. Specifically, the hands were visible, on the sides of the stimuli, and facing the stimuli in the proximal position, all features associated with larger hand proximity effects in the literature [4, 5, 17]. Even with these considerations, however, we did not find an effect of hand proximity.

However, null effects should be interpreted with caution. In particular, it is possible that our experiment was under-powered. We suggest that this is unlikely for a number of reasons. First, our sample was much larger than the one included in Adam and colleagues (43 vs. 20 participants) and based on power analyses. In addition, to ensure power in terms of number of

observations as well, the number of trials per condition matched those used by Adam and colleagues (38 per condition; because Adam et al. had 19 trials in each static and dynamic condition, and therefore averaged 38 trials for their main effect, we doubled the number of trials per condition from 19 to 38). Finally, the Bayes Factor for hand position provided moderate evidence in favor of the null hypothesis.

Our study was specifically aimed to test whether hand proximity affects letter identification in stationary conditions. As such, a number of differences must be noted between our experiment and the original one. For example, accuracy levels were higher in Adam et al.: accuracy of letter identification for left-to-right positions was 92%, 84%, 83.4% in their first experiment, while it was 82.5%, 61.5%, 62.3% in the current study. Adam et al. presented the letters in a white rectangular frame in the middle of the monitor. The current study, on the other hand, used a small fixation point that disappeared at the onset of the target letters. This procedural difference may have altered the attentional focus of the participants, contributing to the observed overall performance level differences. Alternatively, or perhaps in addition, a standing position, adopted by Adam and colleagues, might have enhanced some cognitive abilities compared to the seated position adopted in our study. For example, Smith and colleagues [19] recently found enhanced cognitive control (e.g., disappearance of the Stroop effect) when participants performed the task while standing as opposed to sitting. It is possible that Adam et al.'s participants benefitted from such enhancement, although cognitive control demands were minimal in the letter identification task adopted by Adam et al. in their Experiment 1 and here. Regardless of the reason for the performance differences, we found the same pattern of results in terms of letter position and stimulus duration as Adam et al., suggesting that these patterns are robust.

Two other differences are more relevant for the discrepancy of results pertaining to the hand proximity effect. First, while the participants' hands surrounded the stimulus in the present study, participants in the Adam et al. study always rested their hands on two keypads in both static and dynamic condition. Moreover, in Adam et al., the participant's hands were strapped to these pads with Velcro bands. This arrangement may have increased the intensity of tactile and/or proprioceptive information available at the hands, possibly enhancing the involvement of the bimodal neuron system in letter identification performance. This difference might explain the difference in outcome between the two studies (we thank a reviewer for suggesting this possibility). Second, Adam and colleagues varied the position of the hands in a systematic way (near, intermediate, far) with relatively minor changes in posture (arms/upper body). We adopted two hand positions (near and far) that were associated with different hand/arm postures. The differing body positions between our near and far conditions perhaps made detection of small near-hands effects difficult. We acknowledge that these differences might explain our inability to find hand proximity effects in the present study. However, it is worth nothing that intense tactile and/or proprioceptive information is not necessary for hand proximity effects to be observed [4, 5]; in addition, most studies on hand proximity effects adopted different arm postures for the proximal and distal positions [e.g., 6–10] and some adopted different hand postures [e.g., 4, 9]. Future research should clarify why some features might be critical in some experimental conditions but not in others (e.g., tactile/proprioceptive information, non-visible hands, stimuli presented outside the hands vs. visible hands that faced and surrounded the stimuli).

## Conclusions

In summary, our findings do not support the hypothesis that hand proximity enhances letter identification when the hands are kept static near the stimuli. They provide a more

conservative test than the original study by Adam and colleagues [13], as the specific contribution of hand proximity in static trials was not assessed in that study. In addition, they provide context to interpret Festman et al.'s [14, 15] results. Their near-hands effects were weak or absent, and not corroborated statistically; our findings suggest that they might not have been present. Importantly, in terms of application to real life situations, it is unlikely that hand proximity affects reading during the stages of letter identification. Our results are in line with those by Grossi and colleagues [12], who failed to find near-hands effects on semantic processing during word, sentence, and picture processing. Perhaps the processing of printed letters and words, especially in tasks requiring the retrieval of abstract information (i.e., identity, meaning), does not engage, or engages minimally, the mechanisms underlying near-hands effects, as they are not typically the target of goal-directed actions (the material on which they are printed, such as paper and books, are).

## Supporting information

**S1 Dataset. Data set for the letter identification task.**
(XLSX)

## Acknowledgments

We would like to thank Emilia Lisiecki and Lucinda Judson for their help with data collection. Many thanks to Andrzej Lisiecki as well for the custom-made chinrest used in this study. We also thank the two reviewers for their thoughtful comments.

## Author Contributions

**Conceptualization:** Giordana Grossi.

**Data curation:** Giordana Grossi.

**Formal analysis:** Giordana Grossi.

**Methodology:** Giordana Grossi, Annie J. Olmstead, Danielle Lukaszewski.

**Supervision:** Giordana Grossi.

**Visualization:** Giordana Grossi.

**Writing – original draft:** Giordana Grossi.

**Writing – review & editing:** Giordana Grossi, Annie J. Olmstead, Danielle Lukaszewski.

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
