## [Decision Letter · Decision Letter 0]

4 Dec 2022

PONE-D-22-28697Does hand proximity enhance letter identification?PLOS ONE

Dear Dr. Grossi,

Thank you for submitting your manuscript to PLOS ONE. After careful consideration, we feel that it has merit but does not fully meet PLOS ONE’s publication criteria as it currently stands. Therefore, we invite you to submit a revised version of the manuscript that addresses the points raised during the review process.

ACADEMIC EDITOR:

I appreciate replication studies and I think that science really needs them. Also, in line with the Reviewers, I found your study a solid and rigorous attempt to replicate. However, I would have expected to get more conclusive findings, by a direct test of the explanation given for the failure to replicate (lines 281-283). The question whether the effects reported by Adams et al. are the result of a Type I error, or they are dependent on specific experimental conditions (e.g., moving the hands) remains open. I think that you should either provide further empirical evidence to disentangle these two hypotheses (which would be a consistent improvement towards the understanding of the phenomenon under investigation), or you should acknowledge more clearly this point in the manuscript.

In line with my thoughts, Reviewer 1 (the first author of the original study you tried to replicate)  found your conclusion somehow too strong and asked to smooth it, pointing out that differences in the experimental procedure could account for the replication failure. Reviewer 2 asked to make explicit from the beginning of the manuscript whether your study is powerful enough to support the null hypothesis. In this respect, I am wondering whether the Bayesian t-test you performed on SPSS used informed priors to compute the BF (Rouder et al., 2009).

ROUDER
J.N., SPECKMAN
P.L., SUN
D., MOREY
R.D., IVERSON
G.
(2009). Bayesian t-tests for accepting and rejecting the null hypothesis.
Psychonomic Bulletin & Review,
16, 225-237.

We look forward to receiving your revised manuscript.

Kind regards,

Francesca Peressotti, Ph.D

Academic Editor

PLOS ONE

Journal Requirements:

2. Please note that in order to use the direct billing option the corresponding author must be affiliated with the chosen institute. Please either amend your manuscript to change the affiliation or corresponding author, or email us at plosone@plos.org with a request to remove this option.

4.Please review your reference list to ensure that it is complete and correct. If you have cited papers that have been retracted, please include the rationale for doing so in the manuscript text, or remove these references and replace them with relevant current references. Any changes to the reference list should be mentioned in the rebuttal letter that accompanies your revised manuscript. If you need to cite a retracted article, indicate the article’s retracted status in the References list and also include a citation and full reference for the retraction notice.

Reviewers' comments:

Reviewer's Responses to Questions

**Comments to the Author**

1. Is the manuscript technically sound, and do the data support the conclusions?

Reviewer #1: Yes

Reviewer #2: Yes

2. Has the statistical analysis been performed appropriately and rigorously? 

Reviewer #1: Yes

Reviewer #2: Yes

3. Have the authors made all data underlying the findings in their manuscript fully available?

Reviewer #1: Yes

Reviewer #2: Yes

4. Is the manuscript presented in an intelligible fashion and written in standard English?

Reviewer #1: Yes

Reviewer #2: Yes

5. Review Comments to the Author

Reviewer #1: This well-written paper reports a failed attempt to replicate a previously reported near-hands facilitative effect on letter identification (article by Adam et al., 2012). Even though it is clear that the present authors did their very best to provide a close replication, careful examination of the respective experimental protocols / designs reveals potentially important differences. Moreover, given the plethora of studies that have documented “altered vision near the hands” in all kind of perceptual tasks, I believe the authors could be asked to provide a more elaborate discussion of their null effect. To be clear, the authors do discuss, and reject, several task variables that may have contributed to the discrepant set of findings (including the visibility / orientation of the hands and power issues). Nevertheless, I believe that additional factors may be considered when discussing the divergent outcomes:

1. The authors investigated the static condition only, whereas Adam et al included a static and dynamic condition, each performed on separate days. Hence, the number of trials in the Adam et al study was about twice as large, which may have boosted overall performance level and its consistency. In line with this observation, identification performance was substantially better in the Adam et al study than in the current report, even though the identification task was very similar (report the identity of 3 short-duration letters). Specifically, accuracy of letter identification for left-to-right positions was 92%, 84%, 83% in the Adam study (Exp1), while it was 82%, 62%, 62% in the current study. Hence, a remarkable, overall performance difference of 17%.

2. Adam et al presented the to-be-identified letters in a white rectangular frame in the middle of the monitor. The current study, on the other hand, used a small fixation sign instead that disappeared at onset of the target letters. This procedural difference may have altered the shape/width and thus efficiency of the attentional focus, perhaps also contributing to the observed overall performance level differences.

3. In the Adam study, the participants always rested their hands on two keypads, and not only in the dynamic condition, this was also the case in the static condition. Moreover, and most importantly, the hands were strapped to these pads with Velcro bands. This may have increased the intensity of tactile/proprioceptive information emerging from the hands, possibly enhancing the involvement of the bimodal neuron system in letter identification performance.

4. Adam et al varied the position of the hands in a very systematic way (near, intermediate, far) with relatively minor changes in posture (arms/upper body). The study under review, however, only used 2 hand positions (near and far) that were associated with completely different hand/arm postures, which may have introduced unwanted confounds, possibly clouding the detection of small near-hands effects.

I believe a balanced evaluation of the present report should consider the above factors that, in isolation or combined, may have played a role in the failure to find a near-hands effect.

Reviewer #2: The authors present the results of a single experiment examining the effect of hand proximity on a letter identification task. Participants attempted to identify 3 letters presented at a variety of short durations under hands proximal and hands distal conditions. Whereas previous studies claiming to show evidence for a near-hands effect in letter identification asked participants to move their hands during the task, in the current experiment, the hands remained static during the task. The authors found robust effects of letter duration and position, but found no evidence of a hand proximity effect.

I thought this was a well-written paper presenting a solid experiment. The Introduction appropriately describes the relevant literature and provides a reasonable justification for the current work. The methods and analyses were appropriate and I think readers who are interested in hand proximity effects will be able to draw reasonable conclusions from the data presented. I have only a few minor suggestions that I think could improve the clarity of the manuscript and enhance readers' understanding:

1. As I was reading the Methods and Results sections, I was skeptical about drawing conclusions from a null result of hand proximity. The authors do a nice job in their Discussion of presenting support that their study is adequately powered and that the evidence in favor of the null hypothesis is moderate, but I think presenting this information earlier in the manuscript would be helpful to readers and lower their initial skepticism. I'd recommend describing the power analysis in the Participants section and including the BF in the Results section.

2. Although the information presented in Table 1 is thorough, I also think readers might benefit from the addition of separate lines for the hands proximal and hands distal conditions in Figure 2. A graphical depiction of the extent to which results were similar between the two hands conditions would help illustrate the authors' argument.

6. PLOS authors have the option to publish the peer review history of their article (what does this mean?). If published, this will include your full peer review and any attached files.

Reviewer #1: **Yes: **Jos Adam

Reviewer #2: No

---

## [Author Response · Author response to Decision Letter 0]

11 Jan 2023

Editor

I appreciate replication studies and I think that science really needs them. Also, in line with the Reviewers, I found your study a solid and rigorous attempt to replicate. However, I would have expected to get more conclusive findings, by a direct test of the explanation given for the failure to replicate (lines 281-283). The question whether the effects reported by Adams et al. are the result of a Type I error, or they are dependent on specific experimental conditions (e.g., moving the hands) remains open. I think that you should either provide further empirical evidence to disentangle these two hypotheses (which would be a consistent improvement towards the understanding of the phenomenon under investigation), or you should acknowledge more clearly this point in the manuscript.

Thank you. We have addressed this point in the Discussion section.

In line with my thoughts, Reviewer 1 (the first author of the original study you tried to replicate) found your conclusion somehow too strong and asked to smooth it, pointing out that differences in the experimental procedure could account for the replication failure. Reviewer 2 asked to make explicit from the beginning of the manuscript whether your study is powerful enough to support the null hypothesis. In this respect, I am wondering whether the Bayesian t-test you performed on SPSS used informed priors to compute the BF (Rouder et al., 2009).

ROUDER J.N., SPECKMAN P.L., SUN D., MOREY R.D., IVERSON G. (2009). Bayesian t-tests for accepting and rejecting the null hypothesis. Psychonomic Bulletin & Review, 16, 225-237.

Thank you for this question. We have clarified in the Results (Lines 265 - 268) that we did not use priors.

Reviewer #1: This well-written paper reports a failed attempt to replicate a previously reported near-hands facilitative effect on letter identification (article by Adam et al., 2012). Even though it is clear that the present authors did their very best to provide a close replication, careful examination of the respective experimental protocols / designs reveals potentially important differences. Moreover, given the plethora of studies that have documented “altered vision near the hands” in all kind of perceptual tasks, I believe the authors could be asked to provide a more elaborate discussion of their null effect. To be clear, the authors do discuss, and reject, several task variables that may have contributed to the discrepant set of findings (including the visibility / orientation of the hands and power issues). Nevertheless, I believe that additional factors may be considered when discussing the divergent outcomes:

1. The authors investigated the static condition only, whereas Adam et al included a static and dynamic condition, each performed on separate days. Hence, the number of trials in the Adam et al study was about twice as large, which may have boosted overall performance level and its consistency. In line with this observation, identification performance was substantially better in the Adam et al study than in the current report, even though the identification task was very similar (report the identity of 3 short-duration letters). Specifically, accuracy of letter identification for left-to-right positions was 92%, 84%, 83% in the Adam study (Exp1), while it was 82%, 62%, 62% in the current study. Hence, a remarkable, overall performance difference of 17%.

2. Adam et al presented the to-be-identified letters in a white rectangular frame in the middle of the monitor. The current study, on the other hand, used a small fixation sign instead that disappeared at onset of the target letters. This procedural difference may have altered the shape/width and thus efficiency of the attentional focus, perhaps also contributing to the observed overall performance level differences.

3. In the Adam study, the participants always rested their hands on two keypads, and not only in the dynamic condition, this was also the case in the static condition. Moreover, and most importantly, the hands were strapped to these pads with Velcro bands. This may have increased the intensity of tactile/proprioceptive information emerging from the hands, possibly enhancing the involvement of the bimodal neuron system in letter identification performance.

4. Adam et al varied the position of the hands in a very systematic way (near, intermediate, far) with relatively minor changes in posture (arms/upper body). The study under review, however, only used 2 hand positions (near and far) that were associated with completely different hand/arm postures, which may have introduced unwanted confounds, possibly clouding the detection of small near-hands effects.

I believe a balanced evaluation of the present report should consider the above factors that, in isolation or combined, may have played a role in the failure to find a near-hands effect.

Thank you for your comments. We agree that these procedural differences may have contributed to the differences between our findings and Adam et al. We have included a discussion of these issues in the revised Discussion section (Lines 304 - 342). We have also reworked out discussion to provide a more balanced evaluation of our study, as suggested. Finally, we have included one more reference and citation (#19) to address the Reviewer’s first point (Smith et al., 2019, p. 15). 

Reviewer #2: The authors present the results of a single experiment examining the effect of hand proximity on a letter identification task. Participants attempted to identify 3 letters presented at a variety of short durations under hands proximal and hands distal conditions. Whereas previous studies claiming to show evidence for a near-hands effect in letter identification asked participants to move their hands during the task, in the current experiment, the hands remained static during the task. The authors found robust effects of letter duration and position, but found no evidence of a hand proximity effect.

I thought this was a well-written paper presenting a solid experiment. The Introduction appropriately describes the relevant literature and provides a reasonable justification for the current work. The methods and analyses were appropriate and I think readers who are interested in hand proximity effects will be able to draw reasonable conclusions from the data presented. 

Thank you for your comment.

I have only a few minor suggestions that I think could improve the clarity of the manuscript and enhance readers' understanding:

1. As I was reading the Methods and Results sections, I was skeptical about drawing conclusions from a null result of hand proximity. The authors do a nice job in their Discussion of presenting support that their study is adequately powered and that the evidence in favor of the null hypothesis is moderate, but I think presenting this information earlier in the manuscript would be helpful to readers and lower their initial skepticism. I'd recommend describing the power analysis in the Participants section and including the BF in the Results section.

The two sections have been moved as suggested. The power analysis now appears at the beginning of the Participants section (Lines 190 – 192). The description of the BF analysis now appears in the Results section (Lines 265 – 268). 

2. Although the information presented in Table 1 is thorough, I also think readers might benefit from the addition of separate lines for the hands proximal and hands distal conditions in Figure 2. A graphical depiction of the extent to which results were similar between the two hands conditions would help illustrate the authors' argument.

Thank you for this suggestion. We replaced Figure 2 with a figure showing both hand conditions.

---

## [Editor Report · Decision Letter 1]

13 Jan 2023

Does hand proximity enhance letter identification?

PONE-D-22-28697R1

Dear Dr. Olmstead,

We’re pleased to inform you that your manuscript has been judged scientifically suitable for publication and will be formally accepted for publication once it meets all outstanding technical requirements.

Kind regards,

Francesca Peressotti, Ph.D

Academic Editor

PLOS ONE

---

## [Editor Report · Acceptance letter]

19 Jan 2023

PONE-D-22-28697R1 

Does hand proximity enhance letter identification? 

Dear Dr. Olmstead:

I'm pleased to inform you that your manuscript has been deemed suitable for publication in PLOS ONE. Congratulations! Your manuscript is now with our production department. 

Kind regards, 

on behalf of

Dr. Francesca Peressotti 

Academic Editor

PLOS ONE